# MicroRNA-15a Regulates the Differentiation of Intramuscular Preadipocytes by Targeting *ACAA1*, *ACOX1* and *SCP2* in Chickens

**DOI:** 10.3390/ijms20164063

**Published:** 2019-08-20

**Authors:** Guoxi Li, Shouyi Fu, Yi Chen, Wenjiao Jin, Bin Zhai, Yuanfang Li, Guirong Sun, Ruili Han, Yanbin Wang, Yadong Tian, Hong Li, Xiangtao Kang

**Affiliations:** College of Animal Science and Veterinary Medicine, Henan Agricultural University, Zheng Zhou 450002, Henan, China

**Keywords:** miR-15a, chicken intramuscular preadipocytes, differentiation, *ACAA1*, *ACOX1*, *SCP2*

## Abstract

Our previous studies showed that microRNA-15a (miR-15a) was closely related to intramuscular fat (IMF) deposition in chickens; however, its regulatory mechanism remains unclear. Here, we evaluated the expression characteristics of miR-15a and its relationship with the expression of acetyl-CoA acyltransferase 1 (*ACAA1*), acyl-CoA oxidase 1 (*ACOX1*) and sterol carrier protein 2 (*SCP2*) by qPCR analysis in Gushi chicken breast muscle at 6, 14, 22, and 30 weeks old, where we performed transfection tests of miR-15a mimics in intramuscular preadipocytes and verified the target gene of miR-15a in chicken fibroblasts (DF1). The miR-15a expression level at 30 weeks increased 13.5, 4.5, and 2.7-fold compared with the expression levels at 6, 14, and 22 weeks, respectively. After 6 days of induction, miR-15a over-expression significantly promoted intramuscular adipogenic differentiation and increased cholesterol and triglyceride accumulation in adipocytes. Meanwhile, 48 h after transfection with miR-15a mimics, the expression levels of *ACAA1*, *ACOX1* and *SCP2* genes decreased by 56.52%, 31.18% and 37.14% at the mRNA level in intramuscular preadipocytes. In addition, the co-transfection of miR-15a mimics and *ACAA1*, *ACOX1* and *SCP2* 3′UTR (untranslated region) dual-luciferase vector significantly inhibited dual-luciferase activity in DF1 cells. Taken together, our data demonstrate that miR-15a can reduce fatty acid oxidation by targeting *ACAA1*, *ACOX1*, and *SCP2*, which subsequently indirectly promotes the differentiation of chicken intramuscular preadipocytes.

## 1. Introduction

The chicken (*Gallus gallus*) is an economically important agricultural animal. Intramuscular fat (IMF) is one of the major factors affecting the quality of poultry meat. Therefore, revealing the molecular mechanism of IMF deposition has been a focus of research in the field of poultry genetics and breeding. MicroRNAs (miRNAs) are small endogenous noncoding RNAs that regulate gene expression at the post-transcription level. Some previous studies have confirmed that miRNAs play an important role in abdominal fat development and deposition [1,2,3,4,5], preadipocyte proliferation [6], and IMF deposition [7,8,9] in chickens. Thus, miRNAs have become a major focus of research to explain the post-transcriptional regulation mechanism of IMF deposition and trait formation associated with IMF in poultry. In chickens, many miRNAs have been identified from different tissues and organs or under different physiological conditions. At present, the number of annotated mature miRNAs in the chicken genome has reached 1232 in the miRBase database (Version 22.1, http://www.mirbase.org/, accessed on 1 October 2018). However, studies on the individual function of miRNAs in chicken IMF deposition are quite scarce.

The microRNA-15 (miR-15) family mainly comprises miR-15a, miR-15b, miR-16, miR-195, miR-322, miR-424, miR-457, and miR-497 [10], and these miRNAs are expressed in multiple tissues such as heart, skeletal muscle, liver, kidney, brain, lung, and spleen [11]. As a member of the miR-15 family, miR-15a is known to be highly conserved across species, and its expression is not purely specific to a particular tissue or cell type [10]. Therefore, it has a wide range of functions in many biological contexts [12,13,14,15]. For example, miR-15a as a tumor suppressor can target multiple oncogenes such as *BCL2*, *MCL1*, *CCND1*, and *WNT3A*, and plays a role in cancers such as Chronic Lymphocytic Leukemia (CLL), pituitary adenomas, and prostate carcinoma [16,17,18]. Importantly, a few studies have shown that miR-15a also plays an important role in animal lipid metabolism or adipogenesis [19,20,21,22,23]. Previously, we studied the characterization of miRNA transcriptome profiles related to breast muscle development and IMF deposition in Chinese Gushi chickens and found that miR-15a was significantly upregulated with the development of breast muscle [24]. This suggests that miR-15a may be closely related to IMF deposition in chicken breast muscle. Therefore, miR-15a can be a novel target to reveal the molecular mechanism of IMF deposition in chickens. However, the role of miR-15a in IMF deposition is still poorly understood in chickens.

It is known that the IMF content is mainly linked to the number and size of intramuscular adipocytes [25]. Generally, the number and size of adipocytes is influenced by various cell events such as cell proliferation and differentiation. Previous studies have found that miR-15a regulates cell size and proliferation by fine-tuning delta-like 1 homolog (*DLK1*) in 3T3-L1 preadipocytes [19] and promotes adipogenesis via repressing forkhead box protein O1 (*FoxO1*) in pigs [20]. This suggested that miR-15a may also affect IMF deposition by regulating various adipocyte events such as cell proliferation and differentiation in chickens. Moreover, miRNAs play essential roles in gene expression regulation through binding the 3’-UTR (untranslated region) of their target mRNA [26]. In organisms, one miRNA may act by targeting hundreds of genes. Previously, based on the association analysis of miRNA and mRNA transcriptome profiles, we found that the dynamic expression trend of genes including acetyl-CoA acyltransferase 1 (*ACAA1*), sterol carrier protein 2 (*SCP2*) and acyl-CoA oxidase 1 (*ACOX1*) has a significant negative correlation with the dynamic changes of miR-15a in the breast muscle of Gushi chicken [24]. This suggested that miR-15a may interact with the above genes in chicken breast muscle. Based on these data, we hypothesized that miR-15a may regulate IMF deposition via influencing cell events such as cell proliferation and differentiation by targeting *ACAA1*, *ACOX1* and *SCP2* in chickens. In this study, we verified the targeted relationship between miR-15a and *ACAA1*, *ACOX1* and *SCP2* in intramuscular preadipocytes. These results will contribute to a better understanding of the molecular mechanisms of IMF deposition in the chicken.

## 2. Results

### 2.1. Over-Expression of Mir-15a Promotes the Differentiation of Intramuscular Preadipocytes From Chicken Breast Muscle Tissue

The expression profile of miR-15a was determined in Gushi chicken breast muscle tissues at 6, 14, 22, and 30 weeks old, and its expression level was significantly upregulated with the development of breast muscle (Figure 1). In particular, the expression level of miR-15a at 30 weeks increased 13.5, 4.5, and 2.7-fold compared with the expression levels at 6, 14, and 22 weeks, respectively. In addition, the role of miR-15a was further determined in chicken intramuscular preadipocytes. The expression level of miR-15a increased about 150-fold in the miR-15a mimic group compared with the negative control (NC) group at 48 h post-transfection (Figure 2A). After 6 days of induction, miR-15a over-expression was able to significantly elevate the mRNA level of adipogenic marker genes such as *PPARγ* and *C*/*EBPα* (Figure 2B,C) and significantly increase cholesterol and triglyceride accumulation in adipocytes (Figure 2D,E). Oil Red O staining showed that miR-15a over-expression also promotes the formation of lipid droplets in adipocytes (Figure 2F). The above observations suggested that miR-15a may promote the differentiation of chicken intramuscular preadipocytes.

### 2.2. Over-Expression of Mir-15a Inhibits ACAA1, ACOX1 And SCP2 Gene Expression During Differentiation of Chicken Intramuscular Preadipocytes

To further investigate the mechanism of miR-15a promoting preadipocyte differentiation, we predicted that miR-15a had a potential targeting relationship with *ACAA1*, *ACOX1* and *SCP2* (Figure 3A). The results proved that the trend of miR-15a expression was the opposite to that of *ACAA1*, *ACOX1* and *SCP2* in Gushi chicken breast muscle from 6 weeks to 30 weeks (Figure 3B–D). The correlation coefficients (*R*^2^) of expression level between miR-15a and *ACAA1*, between miR-15a and *ACOX1*, and between miR-15a and *SCP2* were −0.50, −0.45 and −0.83, respectively. Meanwhile, in intramuscular preadipocytes transfected with miR-15a mimics, the expression levels of *ACAA1*, *ACOX1* and *SCP2* genes decreased by 56.52%, 31.18% and 37.14% at the mRNA level compared with the control group (Figure 4), and the protein expression levels of *ACAA1* and *ACOX1* decreased by 45.45% and 30.00% (Figure 5), respectively. This suggested that miR-15a over-expression inhibited the expression of the *ACAA1*, *ACOX1* and *SCP2* genes in adipocytes.

### 2.3. MiR-15a Can Directly Target ACAA1, ACOX1 and SCP2 in Chicken Intramuscular Preadipocytes

To further confirm that *ACAA1*, *ACOX1* and *SCP2* were the target genes of miR-15a, we first determined the expression relationship between miR-15a and these genes in chicken fibroblasts (DF1) transfected with the miR-15a mimic. After 48 h of transfection, the miR-15a expression level increased about 70-fold compared with the NC (Figure 6A). Meanwhile, miR-15a over-expression also significantly inhibited the expression of *ACAA1*, *ACOX1* and *SCP2* in DF1 cells (Figure 6B–D). Subsequently, a luciferase activity assay was performed in DF1 cells. For the *ACAA1* gene, after 48 h of co-transfection, the luciferase activity of the miR-15a group was significantly lower than that of the NC group, and this reduction disappeared in the mutation group (Figure 6E). Likewise, the co-transfection of miR-15a mimics and *ACOX1* 3′UTR dual-luciferase vector significantly inhibited dual-luciferase activity (Figure 6F). For the *SCP2* gene, co-transfection of miR-15a and *SCP2* 3′UTR dual-luciferase reporter vector significantly inhibited the activity of the wild psiCHECK-SCP2-3′UTR reporter, while the mutant reporter vector did not change (Figure 6G).

## 3. Discussion

A few studies of miR-15a have been reported in chickens [27,28]. However, the role of miR-15a in IMF deposition has not been reported in the chicken. Previously, we studied the characterization of miRNA transcriptome profiles related to breast muscle development and IMF deposition in Chinese Gushi chickens and found that miR-15a was significantly upregulated with the development of breast muscle [24]. In particular, the change in the trend of miR-15a expression level was highly consistent with the change in the trend of IMF content in breast muscle. In the present study, we proved the expression characteristics of miR-15a by RT-PCR analysis (Figure 1). This suggests that miR-15a is closely related to IMF deposition in chicken breast muscle. In fact, some studies have demonstrated that miR-15a plays an important role in animal lipid metabolism [21,22,23]. For example, miR-15a represses the expression of low-density lipoprotein receptor-related protein 6 in goats and promotes fat metabolism in goat mammary epithelial cells [21]. Previous studies have shown that miR-15a downregulates fatty acid synthase (FASN) expression in mammary cells [22]. Importantly, literature retrieval found that miR-15a regulates cell size and proliferation in 3T3-L1 preadipocytes [19] and promotes adipogenesis in pigs [20]. In the present study, we found that the over-expression of miR-15a in intramuscular preadipocytes of chicken breast muscle can upregulate the expression of adipogenic marker genes such as *PPARγ* and *C*/*EBPα*, can increase the accumulation of cholesterol and triglycerides, and can promote the formation of lipid droplets in adipocytes (Figure 2). These results are consistent with those reported in 3T3-L1 preadipocytes [19] and porcine preadipocytes [20]. Thus, the results obtained in this study demonstrate that miR-15a may be a facilitator of differentiation of chicken intramuscular preadipocytes and play an important role in IMF deposition in chicken breast muscle.

Previously, we found that miR-15a had potential interactions with the *ACOX1*, *ACAA1*, and *SCP2* genes, based on the results of association analysis of the miRNA and mRNA transcriptome profiles in Gushi chicken breast muscle [24]. In this study, we further verified that the trend of miR-15a expression was the opposite to that of *ACOX1*, *ACAA1*, and *SCP2* in Gushi chicken breast muscle from 6 weeks to 30 weeks by real-time qPCR (Figure 3B–D). In particular, the expression of the *ACOX1*, *ACAA1*, and *SCP2* genes was decreased significantly by miR-15a over-expression in chicken intramuscular preadipocytes (Figure 4 and Figure 5). Subsequently, the luciferase activity assay demonstrated that miR-15a can directly target *ACOX1*, *ACAA1*, and *SCP2* (Figure 6E–G). Altogether, these results indicate that *ACOX1*, *ACAA1*, and *SCP2* are direct target genes of miR-15a in chicken intramuscular preadipocytes. However, the results differ from some previously reported target genes of miR-15a, such as *FoxO1*, *DLK1*, *UCP*-2, and *FASN* [19,20,22,23]. In porcine preadipocytes, miR-15a represses the expression of *FoxO1* [20]; this transcription factor is associated with cell metabolism, proliferation, differentiation, and apoptosis [29]. In 3T3-L1 preadipocytes, miR-15a fine-tunes the level of *DLK1* [19]; this gene is implicated in cellular growth and plays multiple roles in development, tissue regeneration, and cancer. In mammary cells, miR-15a can target *FASN* and downregulate its expression [22]; *FASN* is the central enzyme promoting the de novo synthesis of long-chain fatty acids. This discrepancy in results may be due to differences between species and cell types. Therefore, these results also indicate that the mechanism of miR-15a regulating IMF deposition in chicken may be different from other species, and miR-15a promotes adipogenesis by interacting with multiple target genes in chicken adipocytes.

The common functional features of the *ACOX1*, *ACAA1*, and *SCP2* genes are closely related to the peroxisomal β-oxidation of fatty acids, and they play an important role in some pathways such as fatty acid metabolism, fatty acid degradation, and biosynthesis of unsaturated fatty acids. In peroxisomes, the protein encoded by the *ACOX1* gene acts first; this protein is a rate-limiting enzyme which catalyzes the desaturation of acyl-CoAs to 2-*trans*-enoyl-CoAs [30,31,32,33], and is specific for the desaturation of straight chain CoA esters of very long-chain fatty acids (VLCFAs), dicarboxylic acids, polyunsaturated fatty acids (PUFA), and prostaglandins [34]. *ACOX1* is commonly expressed in most tissues and plays a conserved key role in vertebrate fatty acid metabolism [32]. Furthermore, in the peroxisome, *ACAAl* and *SCP-2* are two thiolase enzymes which catalyze the final stage of beta-oxidation [35]. *ACAA1* encodes an enzyme which cleaves 3-ketoacyl CoA to give acetyl-CoA and acyl-CoA during the fatty acid beta-oxidation cycle which takes place in the peroxisome [36]. *SCP2* has no enzymatic activity but binds branched-chain lipids and enhances branched-chain fatty acid cellular uptake and metabolism [37]. Within the peroxisomal matrix, *SCP2* directly interacts with fatty acid oxidative enzymes and binds branched-chain fatty acyl-CoAs to these enzymes to facilitate their oxidation [38]. *SCP2* is highly expressed in organs involved in lipid metabolism, and its encoded protein is thought to be an intracellular lipid transfer protein. The results of the present study demonstrate that miR-15a can directly target *ACOX1*, *ACAA1*, and *SCP2* in intramuscular preadipocytes of chicken breast muscle (Figure 6E–G). These results, along with those from previous studies, indicate that the role of miR-15a in chicken intramuscular adipocytes is mainly to reduce the beta-oxidation of fatty acids by inhibiting the expression of *ACAA1*, *ACOX1*, and *SCP2* genes in the peroxisomal β-oxidation of fatty acids, thereby promoting IMF deposition in chickens.

It is known that peroxisome proliferator-activated receptors (PPARs) are nuclear hormone receptors activated by fatty acids and their derivatives, and the PPAR signaling pathway is closely associated with biological processes such as lipid metabolism and cell differentiation. We note that the *ACAA1*, *ACOX1*, and *SCP2* genes, which are directly targeted by miR-15a in this study, are downstream target genes of *PPARs* and are under the control of *PPARs* [39]. In the PPAR signaling pathway, unsaturated fatty acids, as ligands, can promote the expression of *PPARs*, and the upregulation of downstream genes such as *ACOX1*, *ACAA1*, *SCP2*, *LCAD*, *CPT-2*, and *CPT-1* by PPAR activation can stimulate fatty acid oxidation. Meanwhile, the upregulation of *aP2* by PPAR activation can also promote adipocyte differentiation. This study confirmed that the over-expression of miR-15a could promote the differentiation of chicken intramuscular preadipocytes (Figure 2). Based on the above results and those of a previous study, we speculate an underlining mechanism as follows: miR-15a reduces the role of fatty acid oxidation in the PPAR signaling pathway by directly targeting *ACAA1*, *ACOX1*, and *SCP2* and inhibiting the expression of these genes, thereby increasing the saturated fatty acid and unsaturated fatty acid content in intramuscular adipocytes of the chicken. The accumulated fatty acids further promote the transcription activity of *PPARs*, and then the expression of downstream adipogenic genes is increased, resulting in the differentiation of chicken intramuscular preadipocytes (Figure 7).

In conclusion, our data indicate that miR-15a can reduce fatty acid oxidation in the PPAR signaling pathway, at least partially, by targeting *ACAA1*, *ACOX1*, and *SCP2*, which subsequently indirectly promotes the differentiation of chicken intramuscular preadipocytes by increasing the accumulation of fatty acids. These findings will provide new molecular insights for understanding the mechanisms of IMF deposition in the chicken and improving meat quality in the practice of animal husbandry. 

## 4. Materials and Methods

### 4.1. Sample Collection

A Chinese domestic breed of Gushi chicken was used in this study. All animal procedures were approved by the Institutional Animal Care and Use Committee (IACUC) of Henan Agricultural University (Use Committee of Henan Agricultural University, China; Permit Number: 17-0118). Healthy female Gushi chickens were selected at the ages of 6, 14, 22, and 30 weeks old, respectively, and 6 individuals were sampled randomly for every developmental stage. The breast muscle tissue was collected immediately after slaughter, snap frozen in liquid nitrogen, and then stored at −80 °C. These samples were used for tissue expression analysis of miR-15a and its predicted target genes.

### 4.2. Isolation, Culture, and Differentiation Induction of Intramuscular Preadipocytes

Gushi chickens were sacrificed 7 days after hatching and the breast muscle tissue was collected immediately. Breast muscle tissue was washed twice with phosphate-buffered saline (PBS) containing 2% double antibody and cut into pieces. The tissue pieces were then digested into floc by a digestion solution containing 0.2% type II collagenase (Invitrogen, Carlsbad, CA, USA) in a 37 °C water bath. After the end of digestion, cells were filtered through a 400-mesh sieve to remove undigested tissue and large cell debris. Subsequently, cells were resuspended in DMEM/F12 medium containing 10% fetal bovine serum (FBS) (Hyclone, Logan, UT, USA) and inoculated into a culture flask at a density of 1 × 10^6^ cells. After the complete medium (DMEM/F12 with 10% FBS, 10,000 U/L double antibody) was added to the culture flask, cells were incubated in a humidified atmosphere of 5% CO_2_ at 37 °C. One hour later, the medium was discarded, and the adherent cells were washed twice with PBS buffer to obtain intramuscular preadipocytes. Finally, preadipocytes were further cultured in complete medium, and the medium was changed once every 2 days.

The induced differentiation of preadipocytes was carried out in inducing medium I (DMEM/F12 with 5% FBS, 0.25 μmol/L dexamethasone and 10 μg/mL insulin). After 2 days of induction, the medium was changed to inducing medium II (DMEM/F12 with 5% FBS and 10 μg/mL insulin) and the culture was continued. Subsequently, the medium was changed once every 2 days.

### 4.3. Transfection of miR-15a Mimics

Chicken intramuscular preadipocytes were seeded in 6-well plates. The cells were transfected when intramuscular preadipocytes grew to 80% confluence. MiR-15a mimics and negative control (NC) were transfected into the cells using Lipofectamine 2000 (Invitrogen, Carlsbad, CA, USA), according to the manufacturer’s protocol. The volume ratio between Lipofectamine 2000 and miR-15a mimics was 1:1. MiR-15a mimics and NC were purchased from Genepharma (Shanghai, China). In addition, intramuscular preadipocytes were induced to differentiation after 6 h transfection, and cells were collected after 6 days of induction to evaluate the changes in relevant indicators.

### 4.4. Construction of Dual-Luciferase Reporter Vector and Luciferase Activity Assay

The partial 3′ UTRs of *ACAA1*, *ACOX1* and *SCP2* containing a miR-15a binding site were amplified from chicken genome DNA by PCR using the primers shown in Table 1, and cloned into the Xho1-Not1 site of the psiCHECK-2 vector (Promega, Maddison, WI, USA), respectively. These vectors were named psiCHECK2-*ACAA1*-3′ UTR-WT, psiCHECK2-*ACOX1*-3′ UTR-WT and psiCHECK2-*SCP2*-3′ UTR-WT, respectively. In addition, the seed region of the miR-15a binding sites was mutated by overlap PCR to generate the mutant 3′ UTR of the *ACAA1*, *ACOX1* and *SCP2* reporters, which were named psiCHECK2-*ACAA1*-3′ UTR-Mut, psiCHECK2-*ACOX1*-3′UTR-Mut and psiCHECK2-*SCP2*-3′ UTR-Mut, respectively. The results of vector construction were confirmed by PCR and sequencing (Sangon Biotech, Shanghai, China). Plasmid DNA was extracted and purified using the EndoFree Maxi Plasmid Kit (Tiangen, Beijing, China).

DF1 cells were seeded in 12-well plates with a density of 1 × 10^5^ cells per well. After 24 h, miR-15a mimics were co-transfected into the cells with a wild-type or mutant-type psiCHECK2-*ACAA1*-vector, psiCHECK2-*ACOX1*-vector and psiCHECK2-*SCP2*-vector, respectively. After 6 h of transfection, the medium was changed. Cells were harvested and lysed after 48 h of transfection. Luciferase activity was measured using the Dual-Luciferase Report Assay System (Promega, Maddison, WI, USA) on a Fluoroskan Ascent FL instrument (Thermo Fisher Scientific, Shanghai, China). The luciferase activity was normalized by the ratio of Renilla luciferase activity and firefly luciferase activity. This transfection experiment was performed in triplicate wells and repeated at least three times.

### 4.5. Triglyceride and Cholesterol Assay

In this study, triglyceride in adipocytes was measured using a Tissue Triglyceride Assay kit (Applygen, Beijing, China), and cholesterol in adipocytes was measured using a Tissue Total Cholesterol Assay kit (Applygen), according to the manufacturer’s instructions.

### 4.6. Oil Red O Staining

Adipogenesis of intramuscular preadipocytes was evaluated by Oil Red O staining and extraction, as previously described [40]. The cultured preadipocytes were gently washed three times with PBS and fixed with 4% paraformaldehyde for 10 min. The fixed cells were washed twice with PBS and stained with 0.5% Oil Red O for 10 min. Then, images of red-stained adipocytes were obtained with a microscope and quantified by examining the spectrophotometric absorbance at 500 nm using a UV-2102 PC ultraviolet spectrophotometer (Unico Instrument Co., Ltd., Shanghai, China).

### 4.7. Quantitative Real-Time PCR (qPCR)

Total RNA of tissue or cell samples was extracted using TRIzol reagent (TaKaRa, Dalian, China), according to the manufacturer’s protocol. The RNA concentrations and integrity were determined by NanoDrop 2000 spectrophotometry (Thermo Scientific, Wilmington, DE, USA) at a ratio of 260/280 nm and assayed by 1.5% agarose gel electrophoresis, respectively. Total RNAs were reverse transcribed using a Prime Script™ RT Reagent Kit (TaKaRa, Dalian, China). The reverse transcription miR-15a stem-loop primer was 5′-GTCGTATCCAGTGCAGGGTCCGAGGTATTCGC

ACTGGATACGACACAAAC-3′, and the reverse transcription U6 stem-loop primer was 5′-GTCGTATCCAGTGCAGGGTCCGAGGTATTCGCACTGGATACGACCGATACA-3′. These stem-loop primers were purchased from GenePharma Co., Ltd. (Shanghai, China). The RT Primer Mix that came with the kit was used for the gene reverse transcription. After reverse transcription, the product was diluted to 400 ng/µL and then stored at −20 °C for later use.

The qPCR was performed using a SYBR Premix Ex Taq II kit (Takara, Dalian, China) on a LightCycler^®^ 96 Real-Time PCR system (Roche, Basel, Switzerland). Three replications were run for every reaction. The reaction system was as follows: 1 µL cDNA product, 5 μL 2× SYBR Premix Ex Taq II, 1 µL specific primer (10 µmol/L), and 3 µL ddH_2_O. The sequences of qRT-PCR primers are shown in Table 2. In addition, the PCR amplification process for miR-15a was as follows: 95 °C for 3 min; 40 cycles of 95 °C for 12 s, 60 °C for 40 s, and 72 °C for 30 s; and 10 min extension at 72 °C. The PCR amplification process for the target genes was as follows: 95 °C for 3 min; 35 cycles of 95 °C for 30 s, 60 °C for 30 s, and 72 °C for 20 s; and 10 min extension at 72 °C. The relative expression levels were calculated using the 2^−ΔΔ*C*t^ method [41]. The U6 small nuclear RNA was used as an endogenous control for miR-15a. In addition, three common chicken housekeeping genes (*GAPDH*, *β-actin*, and *B2M*) were tested prior to target gene quantification [42], and they were evaluated using GeNorm [43]. Since *β-actin* and *GAPDH* displayed a lower M value, these two housekeeping genes was used as references for the target genes.

### 4.8. Western Blot Analysis

The total protein of adipocyte samples was extracted by using RIPA lysis buffer supplemented with phenylmethylsulfonyl fluoride (Servicebio, Wuhan, China) (100:1). Proteins in cell lysates were separated by 10% SDS-PAGE gel electrophoresis and transferred to polyvinylidene difluoride (PVDF) membranes (ISEQ00010, Millipore, Boston, MA, USA). The membranes were blocked with 5% nonfat milk for 1 h and then incubated with mouse anti-*ACAA1* (sc-514051, Santa Cruz, Dallas, TX, USA), and mouse anti-*ACOX1* (sc-517306, Santa Cruz, Dallas, TX, USA) antibodies at 4 °C overnight, followed by incubation with a secondary antibody conjugated with horse radish peroxidase (HRP) (GB23303, Servicebio) for 1 h at room temperature. Mouse anti-actin (Servicebio,) was used as the internal control. Signals were enhanced by ECL Plus (Solarbio, Beijing, China), and images were captured and analyzed by Photoshop CS6 and AlphaView 3.0 (Alpha Innotech, San Jose, CA, USA).

### 4.9. Statistical Analysis

Statistical analyses of all experimental data were carried out using SPSS version 19.0 (IBM, Chicago, IL, USA). Data are expressed as the mean ± SEM. Statistical significance of the data was determined using the *t*-test (unpaired, two-tailed). Values of *p* < 0.05 and *p* < 0.01 were considered as a statistically significant difference and a highly significant difference, respectively. Graphics were drawn using GraphPad Prism 5 software (San Diego, CA, USA). In addition, the correlation coefficient (*R*^2^) of expression level between miR-15a and a corresponding predicted target was identified using Pearson’s correlation analysis.

## Figures and Tables

**Figure 1 ijms-20-04063-f001:**
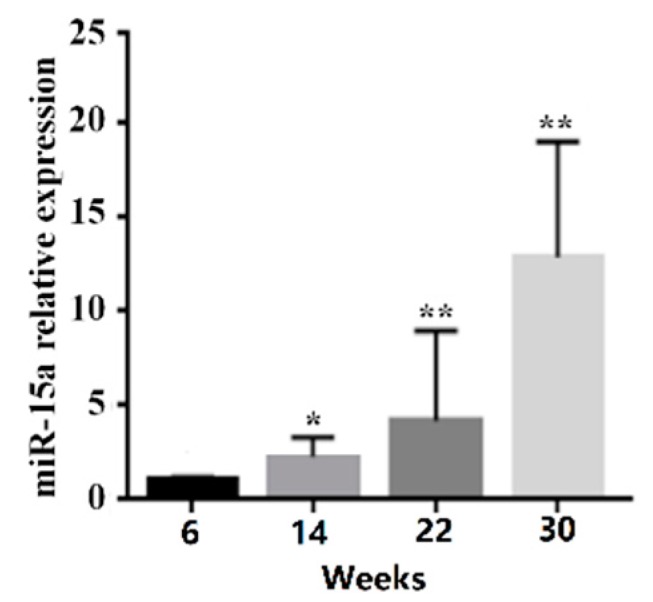
The expression profile of microRNA-15a (miR-15a) in Gushi chicken breast muscle. The numbers 6, 14, 22, and 30 on the X axis denote the samples obtained at 6, 14, 22, and 30 weeks, respectively. Data are expressed as mean ± standard error of mean (SEM) (*n* = 3). * *p* < 0.05, ** *p* < 0.01.

**Figure 2 ijms-20-04063-f002:**
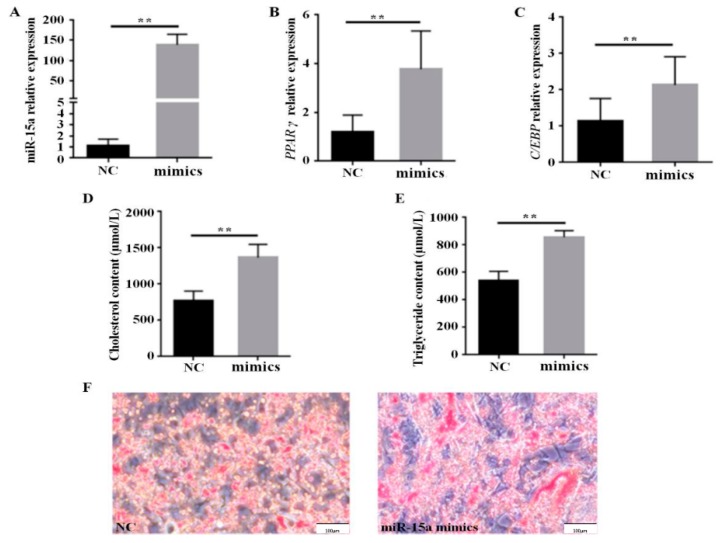
Results of the miR-15a mimic transfection test in chicken intramuscular adipogenesis. (**A**) MiR-15a expression level at 48 h post-transfection; *n* = 3. (**B**) and (**C**) The mRNA level of adipogenic marker genes after 6 days of induction; *n* = 3. (**D**) and (**E**) The cholesterol and triglyceride content assay of differentiated adipocytes after 6 days of induction; *n* = 8. (**F**) Oil Red O staining of differentiated adipocytes after 6 days of induction. Data are represented as the mean ± SEM. ** *p* < 0.01. NC, negative control; mimics, miR-15a mimics.

**Figure 3 ijms-20-04063-f003:**
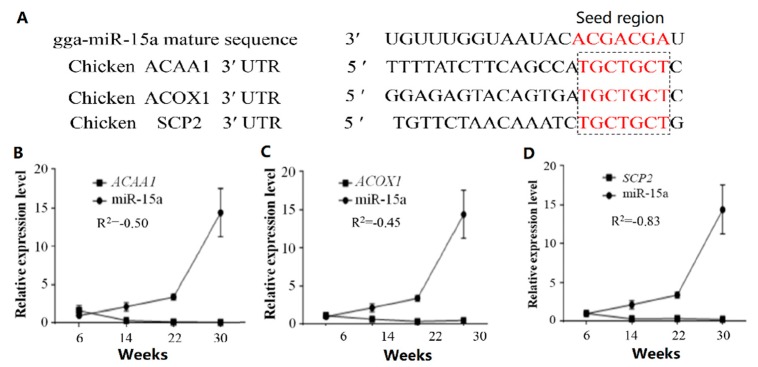
Potential interactions between miR-15a and its predicted target genes. (**A**) The potential binding site of miR-15a in the *ACAA1*, *ACOX1* and *SCP2* mRNA 3′ UTR (untranslated region). (**B**) The relationship of expression level between miR-15a and *ACAA1*. (**C**) The relationship of expression level between miR-15a and *ACOX1*. (**D**) The relationship of expression level between miR-15a and *SCP2*. The numbers 6, 14, 22, and 30 on the X axis denote the samples obtained at 6, 14, 22, and 30 weeks, respectively. *R*^2^ indicates the correlation coefficient of expression level between miR-15a and a corresponding predicted target in the breast muscle of Gushi chicken. Data are expressed as mean ± SEM (*n* = 3).

**Figure 4 ijms-20-04063-f004:**
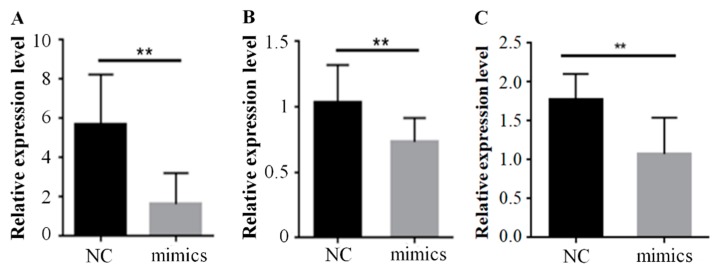
miR-15a over-expression decreased the expression levels of *ACAA1* (**A**), *ACOX1* (**B**) and *SCP2* (**C**) genes at the mRNA level in chicken intramuscular adipocytes. Data are expressed as mean ± SEM (*n* = 3); ** *p* < 0.01. NC, negative control; mimics, miR-15a mimics.

**Figure 5 ijms-20-04063-f005:**
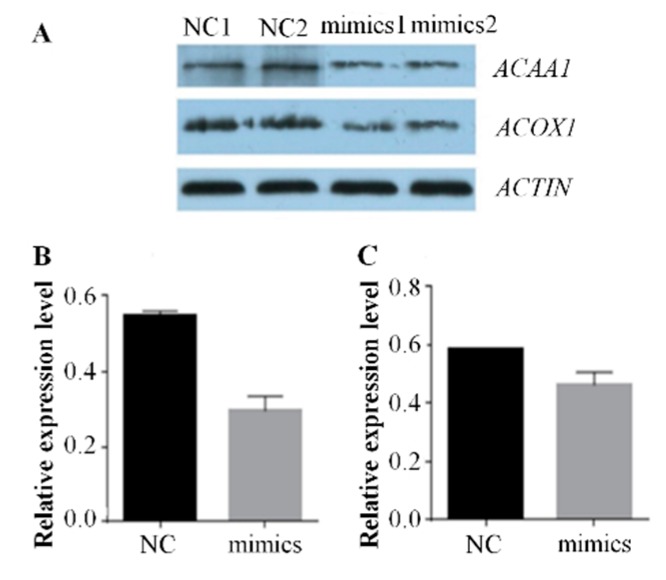
miR-15a over-expression decreased the protein expression levels of *ACAA1* and *ACOX1* genes in chicken intramuscular adipocytes. (**A**) Protein electrophoresis band. (**B**) The relative expression level of *ACAA1* at the protein level. (**C**) The relative expression level of *ACOX1* at the protein level. Data are expressed as mean ± SEM (*n* = 2); * *p* < 0.05, ** *p* < 0.01. NC, negative control; mimics, miR-15a mimics. *ACTIN* was used as a reference gene.

**Figure 6 ijms-20-04063-f006:**
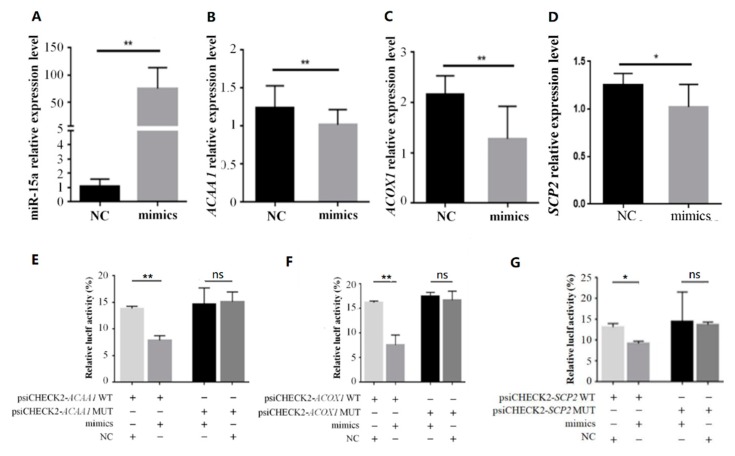
The luciferase assay of miR-15a targeting the 3′UTR of *ACAA1*, *ACOX1* and *SCP2* in DF1 cells. (**A**–**D**) denote the expression levels of miR-15a, *ACAA1*, *ACOX1* and *SCP2* in DF1 cells transfected with miR-15a mimics, respectively. (**E**) Results from tests in which *ACAA1* mRNA 3′ UTR or its mutation in a dual-luciferase vector were co-transfected with miR-15a mimics/NC. (**F**) Results from tests in which *ACOX1* mRNA 3′ UTR or its mutation in a dual-luciferase vector were co-transfected with miR-15a mimics/NC. (**G**) Results from tests in which *SCP2* mRNA 3′ UTR or its mutation in a dual-luciferase vector were co-transfected with miR-15a mimics/NC. The four colored bars in (**E**–**G**) indicate co-transfection tests with four combinations, respectively. Data are expressed as mean ± SEM (*n* = 3); * *p* < 0.05; ** *p* < 0.01; ns, not significant. NC, negative control; mimics, miR-15a mimics.

**Figure 7 ijms-20-04063-f007:**
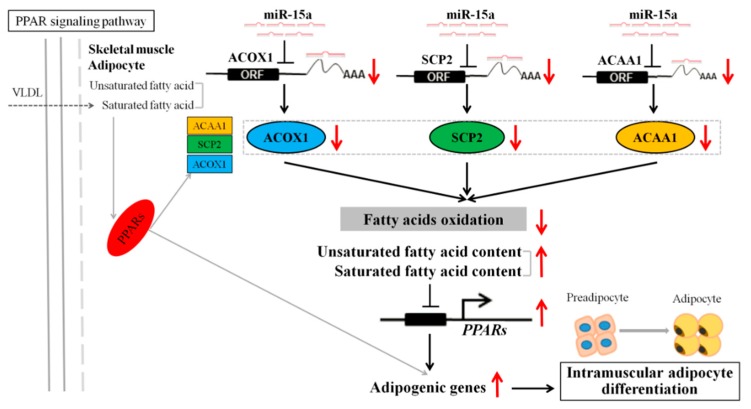
Diagram illustrating the underlying mechanism by which that miR-15a regulates the differentiation of chicken intramuscular preadipocytes by targeting *ACOX1*, *SCP2* and *ACAA1* in the PPAR signaling pathway. A red upward arrow indicates that this process is promoted, or this indicator is increased; a red downward arrow indicates that this process is inhibited, or this indicator is decreased. *PPARs*, peroxisome proliferator-activated receptors; *VLDL*, very low-density lipoprotein; *ACAA1*, acetyl-CoA acyltransferase 1; *SCP2*, sterol carrier protein 2; *ACOX1*, acyl-CoA oxidase 1.

**Table 1 ijms-20-04063-t001:** Primer sequences used for dual-luciferase reporter vector construction.

Primer Name	Primer Sequences (5′–3′)
*ACAA1*-WT	F: CCCTCGAGGCAGCTTGGCAAATGTCTTA
	R: ATTTGCGGCCGCATTCAGGCATCCCAACAGTC
*ACAA1*-ovelap	F: AGCCACTCAGACTTCCAGTGGGG
	R: CCCCACTGGAAGTCTGAGTGGCT
*ACOX1*-WT	F: CCCTCGAG CTTTCACTGCCCTGCAGAAG
	R: ATTTGCGGCCGCAACAGTTAAAAGGGCAGAAAATC
*ACOX1*-ovelap	F: GTACAGTGACTGAGGACA
	R: TGTCCTCAGTCACTGTAC
*SCP2*-WT	F: CCCTCGAGCCTCAGACAGCTCCTTGCTC
	R: ATTTGCGGCCGC TCTGGAAAAGTGGTGGGTTC
*SCP2*-ovelap	F: AATGTTGTCACCGGTATTG
	R: CAATACCGGTGACAACATT

**Table 2 ijms-20-04063-t002:** Primer sequences used for qRT-PCR analysis.

Primer Name	Primer Sequence (5′–3′)	Location	Product Length (bp)	Annealing Temperature (°C)	Accession Number
*ACAA1*	F: CCAGCATACTGACAGCCCAA	1316-1335	170	59	NM_001197288.1
	R: TCCCACTTGCACATCAGACC	1466-1485			
*ACOX1*	F: TTAATGACCCTGACTTCCAGC	193-233	162	58	NM_001006205.1
	R: CACGATGAACAAAGCTTTTAAACCA	330-354			
*SCP2*	F: AGGAGGCAACCTGGGTAGT	1416-1434	159	59	NM_001305200.1
	R: ATTTGCCTTGAAAGAAGGCTGTC	1552-1574			
*PPAR γ*	F: CTCCTTCTCCTCCCTATTT	245-263	227	60	NM_001001460.1
	R: TTTCTTATGGATGCGACA	454-471			
*C*/*EBPα*	F: GACAAGAACAGCAACGAGTACCGC	503-536	195	56	NM_001031459.1
	R: CCTGAAGATGCCCCGCAGAGT	677-697			
*β-actin*	F: GATATTGCTGCGCTCGTTG	78-96	453	60	NM_205518.1
	R: GTCCATCACAATACCAGTGG	511-530			
*GADPH*	F: TGATGGTCCACATGGCATCC	1031-1050	141	60	NM_204305.1
	R: GGGAACAGAACTGGCCTCTC	1152-1171			

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
