# Peer review of "MicroRNA-15a Regulates the Differentiation of Intramuscular Preadipocytes by Targeting ACAA1, ACOX1 and SCP2 in Chickens"

_ijms, 2019, doi:10.3390/ijms20164063_

Round 1

Reviewer 1 Report

The paper by Li is an interesting study on the role of MicroRNA-15 in regulating the differentiation of adipose tissue. Although the topic is interesting, this reviewer has severe concerns on the methods adopted for RTqPCR. Furthermore, the paper should be largely re-written, as it does not reflect the standard structure for a research article in some points.

SPECIFIC COMMENTS:

ABSTRACT:

Please spell out all the acronyms used in the abstract at first use (i.e. IMF, miR15-a). I think that might be also the case to spell out all the genes names at first use.

You didn’t mention anything about your experimental design in the abstract (i.e. number of animals, analysis, statistics). You didn’t report any of your results neither. You only discuss them briefly, while the abstract should give a clear idea of the experimental design adopted and of the results obtained to the reader. Please re-write this whole section.

INTRODUCTION

The introduction is quite long, and I think that should be re-written focusing on the main aims of the research. Many of the information included in the Results and Discussion sections should be moved to the introduction, as they represent a scientific background that is pivotal for the reader to understand your experimental design, your results and the interpretation you gave to them.

MATHERIAL AND METHODS

A main concern to me is the method that you adopted to perform RTqPCR and how you choose the reference gene. The standard methods to perform RTqPCR assay has been deeply described by https://www.ncbi.nlm.nih.gov/m/pubmed/19246619/, while for the choice of reference genes and for normalization of data you should refer to https://genomebiology.biomedcentral.com/articles/10.1186/gb-2002-3-7-research0034 . It has been recently demonstrated that performing RTqPCR without checking for the stability of reference genes with a provided software (i.e. GeNorm, Norm Finder or Best Keeper) will lead to biased results. For details see https://doi.org/10.1016/j.gene.2018.100003 . It seems to me that you choose beta-actin as a reference gene without using any software to checking stability, and thus your results could have been altered from such a choice.

L359: Provide details of antibodies used

L415: Please provide the RIN and the concentration obtained (see https://www.ncbi.nlm.nih.gov/m/pubmed/19246619/)

L444: Define HRP at first use

L448 Please, provide the model that you adopted and explain how did you check the normality assumption

RESULTS

Results section is confused. It includes sub-sections titles that fits more with the discussion section for me: remove all of them or rephrase them in order to avoid any speculation on the results. Results section should only present the results that you obtained in the study, in an objective manner and without any interpretation of them or any personal comment. It seems to me that you mixed the results and discussion sections at some point, and that you also provided some information that should be moved to the M&M section (see comments below).

Figure: all the figures caption should be totally objective, reporting only information that are necessary to the reader to understand the figure. No interpretation of results or methods information should be provided in figure caption!

L108-109: Unnecessary. You have to say that in M&M.

L109: remove “The results showed that”

L110: Be objective: Remove “obvious” (personal comment). Furthermore, it is unclear to me what you mean with “temporal characteristics”

L114: Why you didn’t show them?

L 114-115: Move to the discussion section

Figure 1, 3 and 9: Remove W after each time point on the X axe. Leave only the numbers and write “Weeks” as X axe title.

L121-124: Move to the M&M section

L 125: move to the discussion

L125-127: Move to the M&M section

L 127-128 (up to marker genes): Move to the discussion

L128: αP2 is not significant

L131-133: Move to the discussion

L145-150: Move to the discussion and to the M&M sections

L150: There is no mention to the Pearson correlation coefficient calculation in your statistical analysis section

L152-153: Move to the discussion

L156-157: Move to the discussion

L173-180; 187-189 and 200-201: Move to M&M

L190-191; 192-195; 25-207: Move to the discussion

DISCUSSION

The discussion section does not give any real explanation of your results (except for lines from 288 to 291). It seems to me a literature review that could be partially re-utilized in the introduction part. This whole section should be re-written moving information that are unnecessary from the results section. In the whole discussion section, you have to discuss your results only, avoiding any speculation on results from others.

L216-226; 236-259: Move to the introduction section

L223-225: These information are unfocused on your results. Remove

L232: Need reference

L274: Define PUFAs at first use

L286: What does “Figure 9; unpublished” mean? I don’t get the utility of such a figure anyway

L306: remove question from discussion

L310: Be specific: which PPAR are you referring to?

Figure 10: Move lines 326 to 335 in the text. Leave lines 335 to 338 only in figure caption.

Rewrite conclusions basing on your data after you addressed results and discussion sections.

Author Response

Dear,Thank you very much for working our paper.

We have thoroughly revised our manuscript according to reviewer(s)' comments. In order to facilitate your examination, any changes which we made to the original manuscript is highlighted in red in the revised manuscript.

Comments and Suggestions for Authors

The paper by Li is an interesting study on the role of MicroRNA-15 in regulating the differentiation of adipose tissue. Although the topic is interesting, this reviewer has severe concerns on the methods adopted for RTqPCR. Furthermore, the paper should be largely re-written, as it does not reflect the standard structure for a research article in some points.

Response: Thank you very much for your objective evaluation.

We have thoroughly revised our manuscript according to your comments. At the same time, we also make substantial amendments to our manuscript in the following aspects:

1. The revised manuscript has been sent to a professional English editing service provided by MDPI for the English language editing.

2. The abstract, introduction, results, and discussion section in text have been re-written to make them more reasonable.

3. The contents in Table1 and Table 2 have been perfected.

4. The figures in the text are adjusted, leaving only seven figures. The main changes are as follows:

(1) Figures 6 in the original manuscript has been deleted, and the relevant content has been fused into figure 3 in the revised manuscript.

(2) Figures 7 and Figures 8 in the original manuscript have been merged into figure 6 in the revised manuscript.

(3) The content related to Figures 9 in the original manuscript have been fused into figure 3, figure 4 and figure 6 in the revised manuscript, respectively.

5. Fourteen references have been deleted. Meanwhile, four references have been supplemented. The reference list and citation format in the text have been modified.

Please review it again and give your valuable comments.

SPECIFIC COMMENTS:

ABSTRACT:

Please spell out all the acronyms used in the abstract at first use (i.e. IMF, miR15-a). I think that might be also the case to spell out all the genes names at first use.

You didn’t mention anything about your experimental design in the abstract (i.e. number of animals, analysis, statistics). You didn’t report any of your results neither. You only discuss them briefly, while the abstract should give a clear idea of the experimental design adopted and of the results obtained to the reader. Please re-write this whole section.

Response: We have rewritten the whole abstract section according to your comments. See the "Abstract" section in the revised text.

INTRODUCTION

The introduction is quite long, and I think that should be re-written focusing on the main aims of the research. Many of the information included in the Results and Discussion sections should be moved to the introduction, as they represent a scientific background that is pivotal for the reader to understand your experimental design, your results and the interpretation you gave to them.

Response: We have rewritten the whole introduction section according to your comments. See the "1. Introduction" section in the revised text.

MATHERIAL AND METHODS

(1) A main concern to me is the method that you adopted to perform RTqPCR and how you choose the reference gene. The standard methods to perform RTqPCR assay has been deeply described by https://www.ncbi.nlm.nih.gov/m/pubmed/19246619/, while for the choice of reference genes and for normalization of data you should refer to https://genomebiology.biomedcentral.com/articles/10.1186/gb-2002-3-7-research0034 . It has been recently demonstrated that performing RTqPCR without checking for the stability of reference genes with a provided software (i.e. GeNorm, Norm Finder or Best Keeper) will lead to biased results. For details see https://doi.org/10.1016/j.gene.2018.100003 . It seems to me that you choose beta-actin as a reference gene without using any software to checking stability, and thus your results could have been altered from such a choice.

Response: Reviewer 2 also mentioned this concern. Take your comments into consideration, we have reassessed three common chicken housekeeping genes (GAPDH, β-actin, and B2M) using GeNorm. Since β-actin and GAPDH displayed a lower M value (Table1), we have redone the qRT-PCR experiments using two housekeeping genes, β-actin and GAPDH. Meanwhile, we have also supplemented the method that how to select reference genes in the materials and methods section and two references in the reference list. See the line 352-355 and 494-498 in the revised text.

Table1 Internal reference gene M values were evaluated by GeNorm

Housekeeping genes

M value

β-actin

0.6087

B2M

0.6821

GAPDH

0.6087

(2) L359: Provide details of antibodies used

Response: We have provided details of antibodies. See the line 362-363 in the revised text.

(3) L415: Please provide the RIN and the concentration obtained (see https://www.ncbi.nlm.nih.gov/m/pubmed/19246619/)

Response: We have added relevant information. See the line 333-334 in the revised text.

(4) L444: Define HRP at first use

Response: We have define HRP. See the line 365 in the revised text.

(5) L448 Please, provide the model that you adopted and explain how did you check the normality assumption

Response: We have provided the model. See the line 371-372 in the revised text.

In addition, three replications were run every reaction in the qPCR analysis, which is a small sample. Therefore, the normality assumption cannot be performed. Now, a lot of research literature does this.

RESULTS

(1) Results section is confused. It includes sub-sections titles that fits more with the discussion section for me: remove all of them or rephrase them in order to avoid any speculation on the results. Results section should only present the results that you obtained in the study, in an objective manner and without any interpretation of them or any personal comment. It seems to me that you mixed the results and discussion sections at some point, and that you also provided some information that should be moved to the M&M section (see comments below).

Response: We agree with your comments and have rewritten the whole results section according to your comments. See the "2. Results" section in the revised text.

(2) Figure: all the figures caption should be totally objective, reporting only information that are necessary to the reader to understand the figure. No interpretation of results or methods information should be provided in figure caption!

Response: We have adjusted the figures in the text and revised all the figures caption and legend. See all the figures caption and legend in the revised text.

(3) L108-109: Unnecessary. You have to say that in M&M.

Response: We have removed this sentence according to your comments. See the line 86 in the revised text.

(4) L109: remove “The results showed that”

Response: We have removed “The results showed that” according to your comments. See the line 87 in the revised text.

(5) L110: Be objective: Remove “obvious” (personal comment). Furthermore, it is unclear to me what you mean with “temporal characteristics”

Response: We have changed narration mode. See the line 87-88 in the revised text.

(6) L114: Why you didn’t show them?

Response: This result has been used in other articles (Refs. 24 in the reference list of the revised text). This is only the basis of comparative analysis. In the revised text, we have moved this sentence to the discussion section. See the line 172-173 in the revised text.

 (7) L 114-115: Move to the discussion section

Response: We have moved this sentence to the discussion section according to your comments. See the line 174-175 in the revised text.

(8) Figure 1, 3 and 9: Remove W after each time point on the X axe. Leave only the numbers and write “Weeks” as X axe title.

Response: We have revised them according to your comments. See Figure 1 and 3 in the revised text.

(9) L121-124: Move to the M&M section  

Response: We have deleted these sentences. See the revised text.

(10) L 125: move to the discussion

Response: We have deleted this sentences. See the revised text.

(11) L125-127: Move to the M&M section

Response: We have deleted this sentences. See the revised text.

(12) L 127-128 (up to marker genes): Move to the discussion

Response: We have removed the content. See the revised text.

(13) L128: αP2 is not significant

Response: We have deleted this result. See Figure 1 in the revised text.

(14) L131-133: Move to the discussion

Response: We have moved this sentence to the discussion section according to your comments. See the line 186-188 in the revised text.

(15) L145-150: Move to the discussion and to the M&M sections

Response: We have deleted some content. See the line 112-115 in the revised text.

(16) L150: There is no mention to the Pearson correlation coefficient calculation in your statistical analysis section

Response: We have supplemented it. See the line 374-376 in the revised text.

(17) L152-153: Move to the discussion

Response: We have deleted this content. See the revised text.

(18) L156-157: Move to the discussion

Response: We have deleted this sentence. See the revised text.

(19) L173-180; 187-189 and 200-201: Move to M&M

Response: We have deleted these content. See the revised text.

(20) L190-191; 192-195; 205-207: Move to the discussion

Response: We have deleted these content. See the revised text.

DISCUSSION

(1) The discussion section does not give any real explanation of your results (except for lines from 288 to 291). It seems to me a literature review that could be partially re-utilized in the introduction part. This whole section should be re-written moving information that are unnecessary from the results section. In the whole discussion section, you have to discuss your results only, avoiding any speculation on results from others.

Response: We have rewritten the whole discussion section according to your comments. See the "3. Discussion" section in the revised text.

(2) L216-226; 236-259: Move to the introduction section

Response: Some content have been moved to the introduction section according to your comments, and other content have been deleted. See the revised text.

(3) L223-225: These information are unfocused on your results. Remove

Response: We have deleted this sentence according to your comments. See the revised text.

(4) L232: Need reference

Response: This part is the result of this study, so no references are needed. We have perfected the narrative. See the line 181-185 in the revised text.

(5) L274: Define PUFAs at first use

Response: We have define PUFAs. See the line 215 in the revised text.

(6) L286: What does “Figure 9; unpublished” mean? I don’t get the utility of such a figure anyway

Response: This part is the result of our experiment and related to SCP2 genes. Since specific antibody to SCP2 were not purchased, we only evaluated its expression at mRNA level. So, these results are put into the discussion section as supporting data. As you say, it is not proper to do so. Therefore, when the manuscript was revised, the content related to Figures 9 in the original manuscript has been incorporated into the results section. See the figure 3, figure 4 and figure 6 in the revised text.

 (7) L306: remove question from discussion

Response: We have deleted this sentence according to your comments. See the line 231 in the revised text.

(8) L310: Be specific: which PPAR are you referring to?

Response: This place represents genes. We have changed "PPAR" to "PPARs". See the line 235 in the revised text.

 (9) Figure 10: Move lines 326 to 335 in the text. Leave lines 335 to 338 only in figure caption.

Response: We have revised it according to your comments. See the line 249-255 in the revised text.

(10) Rewrite conclusions basing on your data after you addressed results and discussion sections.

Response: We have rewritten conclusions according to your comments. See the line 256-261 in the revised text.

Reviewer 2 Report

On the whole the manuscript is well-written and clearly understandable in introduction, material and methods and results section. Some parts in discussion section must be checked for english.

Validation using only one housekeeping gene is highly inaccurate. I suggest the Authors read this interesting paper: Joanne R. Chapman and Jonas Waldenström; With Reference to Reference Genes: A Systematic Review of Endogenous Controls in Gene Expression Studies. Published on PLoS One, in 2015; 10(11): e0141853. doi:  10.1371/journal.pone.0141853. Is it possible to add another housekeeping gene? Have the Authors tested more than one reference gene?

Furthermore, concerning Table 2, I strongly advise the authors add the location of primer sequences in the genes (exons? UTRs?), le length of the amplicon, the annealing temperatures (following MiQE guidelines).

I also ask if it is possible for the authors to upload the complete images of protein wstern blot (as supplementary material).

Minor changes:

Lines 154-157: improper use of italics. Please revise.

Figure 8: it is not clear to me what are the 4 bars in the two graphs. Please add a description of the four colored bars.

Line 220: “to a particular tissue or cell types” should be “to a particular tissue or cell type”

Lines 237-238: “differ among species and tissue and cell types and physiological/pathological conditions” maybe too many “and”.

Lines 246-248: “In mammary cells, miR-15a can target FASN and downregulated the expression of FASN [36], FASN is the central enzyme promoting the de novo synthesis of long-chain fatty acids.” I suggest the authors reword this sentence as follows: “In mammary cells, miR-15a can target FASN and downregulate its expression [36]. FASN is the central enzyme promoting the de novo synthesis of long-chain fatty acids.”

Author Response

Dear,Thank you very much for working our paper.

We have thoroughly revised our manuscript according to reviewer(s)' comments. In order to facilitate your examination, any changes which we made to the original manuscript is highlighted in red in the revised manuscript.

Comments and Suggestions for Authors

(1) On the whole the manuscript is well-written and clearly understandable in introduction, material and methods and results section. Some parts in discussion section must be checked for english.

Response: Thank you very much for your objective evaluation.

We have revised our manuscript according to your comments. At the same time, we also make substantial amendments to our manuscript in the following aspects:

1. The revised manuscript has been sent to a professional English editing service provided by MDPI for the English language editing.

2. The abstract, introduction, results, and discussion section in text have been re-written to make them more reasonable.

3. The contents in Table1 and Table 2 have been perfected.

4. The figures in the text are adjusted, leaving only seven figures. The main changes are as follows:

(1) Figures 6 in the original manuscript has been deleted, and the relevant content has been fused into figure 3 in the revised manuscript.

(2) Figures 7 and Figures 8 in the original manuscript have been merged into figure 6 in the revised manuscript.

(3) The content related to Figures 9 in the original manuscript have been fused into figure 3, figure 4 and figure 6 in the revised manuscript, respectively.

5. Fourteen references have been deleted. Meanwhile, four references have been supplemented. The reference list and citation format in the text have been modified.

Please review it again and give your valuable comments.

(2) Validation using only one housekeeping gene is highly inaccurate. I suggest the Authors read this interesting paper: Joanne R. Chapman and Jonas Waldenström; With Reference to Reference Genes: A Systematic Review of Endogenous Controls in Gene Expression Studies. Published on PLoS One, in 2015; 10(11): e0141853. doi:  10.1371/journal.pone.0141853. Is it possible to add another housekeeping gene? Have the Authors tested more than one reference gene?

Response: Reviewer 1 also mentioned this concern. Take your comments into consideration, we have reassessed three common chicken housekeeping genes (GAPDH, β-actin, and B2M) using GeNorm. Since β-actin and GAPDH displayed a lower M value (Table1), we have redone the qRT-PCR experiments using two housekeeping genes, β-actin and GAPDH. Meanwhile, we have also supplemented the method that how to select reference genes in the materials and methods section and two references in the reference list. See the line 352-355 and 494-498 in the revised text.

Table1 Internal reference gene M values were evaluated by GeNorm

Housekeeping genes

M value

β-actin

0.6087

B2M

0.6821

GAPDH

0.6087

(3) Furthermore, concerning Table 2, I strongly advise the authors add the location of primer sequences in the genes (exons? UTRs?), le length of the amplicon, the annealing temperatures (following MiQE guidelines).

Response: We have supplemented relevant information according to your comments. See Table 2 in the revised text.

(4) I also ask if it is possible for the authors to upload the complete images of protein wstern blot (as supplementary material).

Response: We feel very sorry. Since specific antibody to SCP2 gene were not purchased, we only evaluated the expression of SCP2 gene at mRNA level. Therefore, we cannot upload the images of protein wstern blot for the SCP2 gene.

Minor changes:

(1) Lines 154-157: improper use of italics. Please revise.

Response: We have revised them according to your comments. See the line 118-122 in the revised text.

(2) Figure 8: it is not clear to me what are the 4 bars in the two graphs. Please add a description of the four colored bars.

Response: The 4 bars represent the relative activity of dual-luciferase in different co-transfection test, respectively. The four colored bars indicate co-transfection test with four combinations, respectively. We have added a description of the four colored bars according to your comments. See the Figure 6 (line 164-165) in the revised text.

(3) Line 220: “to a particular tissue or cell types” should be “to a particular tissue or cell type”

Response: We have revised it according to your comments. See the line 53 in the revised text.

(4) Lines 237-238: “differ among species and tissue and cell types and physiological/pathological conditions” maybe too many “and”.

Response: We have deleted this sentence. See the revised text.

(5) Lines 246-248: “In mammary cells, miR-15a can target FASN and downregulated the expression of FASN [36], FASN is the central enzyme promoting the de novo synthesis of long-chain fatty acids.” I suggest the authors reword this sentence as follows: “In mammary cells, miR-15a can target FASN and downregulate its expression [36]. FASN is the central enzyme promoting the de novo synthesis of long-chain fatty acids.”

Response: We have revised it according to your comments. See the line 203-205 in the revised text.

Round 2

Reviewer 1 Report

The paper has been improved substantially, but you still need to work on the presentation of your results. Please see the comments below:

ABSTRACT:
Your abstract has been improved substantially, but I still thinking that it needs a further effort at some points. The abstract should include a short introduction presenting the topic of the paper (1 sentence is enough). Then you should clearly state the aim of your study and the hypothesis that you want to test (this have not been included in your abstract). Then, a clear explanation of the experimental design adopted (including the number of animals included in the trial, all the assays performed, and the statistical analysis adopted) should be included. Then you should present your results and include one statement of conclusions (I think that both these sections are quite well written in your abstract). Specific comments are stated below. Please, modify your abstract accordingly and referring to the general structure described above.
L16: Eliminate “Our previous studies showed that”. This kind of sentences should be supported by a reference, and you should make your abstract unpersonal. If these are published data, you could take them as a matter of fact in the abstract. My personal suggestion is to change the sentence to “Micro RNA-15a (miR-15a) is closely related to intramuscular fat (IMF) deposition in chickens; however…” or something similar.
L18: make the abstract unpersonal. Avoid “we” (use passive form)
L18-22: the experimental design you adopted is still unclear and need a further explanation. At L 29-30 you present the results on dual-luciferase, but there is no mention to the dual-luciferase assessment in the description of the experimental design.
L 24: Please, indicate P-values
INTRODUCTION:
Your introduction has been improved substantially. Specific comments are listed below.
L 36-37. Delete the first two sentences. Unnecessary for me (personal comment).
L54-56. Delete. Although interesting, anti-carcinogenic function of miRNA are unfocused with the aim of your study.
MATERIALS AND METHODS:
L 272: Move the sentence at L 267 (After “…developmental age”)
L334. Please provide the 260/280 and the RIN (RNA Integrity Number) values you found
RESULTS:
Your results have been improved substantially, but you still need to work on data presentation. Please, use the comments I provided for the first sub-section as a guide for your results. Furthermore, as a general comment, what does n=3 or n=8 means in figure captions? If they are the number of replicates, for each assay performed you need to define them in the material and methods section too.
L84. As I said in the previous revision, sub-titles in the results section should not be used to discuss the results. Do not anticipate the results that you found in the sub-title (i.e change “Over-expression of miR-15a promotes the differentiation of intramuscular preadipocytes from chicken breast muscle” to “Expression level of miR-15a in breast muscle and intramuscular preadipocytes” or something similar). Please, rephrase all the sub-titles in the results section accordingly.
L 86-88. Please, rephrase the sentence to make it compliant with the results section. (i.e. “The expression level of miR-15a found in breast muscle at 14; 22 and 30 weeks of development was upregulated compared to the level found at 6 weeks (Figure 1)” or something similar)
L 88. Here you present the fold increase calculated on the expression level at 30 weeks, while statistical significances in the figure 1 are referred to the comparison between the expression found at 6 weeks and subsequent observations. Please clarify which time point you adopted as the reference time point for comparisons, and then be consistent both in results presentation and in the figure adopting the same time point to make comparisons (see following comment).
Figure 1: Please indicate which time point you used to make comparisons in figure caption too (i.e. “Fold variations of the microRNA-15a (miR-15a) expression level found in Gushi chicken breast muscle at 14; 22 and 30 weeks of development in respect with the level found at 6 weeks of development. Data are expressed…” or something similar).
L90. Delete sentence (this is not Material and Methods). Change to “In intramuscular preadipocytes the expression level of…”
L93. You cannot say that “miR-15a over-expression was able to significantly elevate the mRNA level of adipogenic marker genes” in the results section (keep that for discussion). I suggest you say only that miR-15a group had higher levels of PPARgamma and C/EBPalpha, and greater cholesterol and triglyceride accumulation in comparison to control group, and to support that with statistics.
L95. Same comment as above. Change to “miR-15a group showed a brighter coloration at Oil Red O staining (Figure 2)” or something similar. Figure 2(F) is unclear anyway, and you need to specify in figure caption which group correspond to which figure (footnotes at the bottom of the image are difficult to read). Finally, is there any numerical value that reflects the results of Oil Red O staining? It could be interesting to quantify the different response to the Oil Red in order to make some statistics (the latest part should be intended as a personal comment).
L97-98. Delete. This is not the discussion section.
L 110-111. See previous comments and change the sub-title accordingly
L112-114. Both these lines, together with figure 3 (A), cannot be considered as a part of your results to me. It should be considered as a part of the introduction (it is a part of the background that drive you to the hypothesis you tested, isn’t it?). I suggest you moving this part to the introduction.
L114-121 Rephrase according with previous comments.
Figure 4 and 5. Do not use figure caption to present your results. See previous comments and rephrase figure caption accordingly.
L121-122. Delete. This is not the discussion section
L143. Rephrase according with previous comments
L144-146 Delete. You already said that in the M&M section
Figure 6. L159. What about negative controls in figure caption?
L147-149. Rephrase according with previous comments
L149. From “Subsequently”to “cells”, Delete. See previous comments
Figure 6 E, F and G. Specify in caption that WT is wild type and MUT is mutate.
DISCUSSION
The discussion is still unfocused on your results. L 168 to 181 are a repetition of what you already said in the introduction and do not discuss any of your results. L 181 to 185 are a repetition of your results, that do not really provide any explanation to them. You should merge these two paragraphs using your results to select the information that are essential to support your interpretation of results. Please do that in the whole discussion section.

Author Response

Dear reviewer,Thank you very much for working our paper.

We have thoroughly revised our manuscript according to your comments. In order to facilitate your examination, any changes which we made to the original manuscript is highlighted in red in the revised manuscript.

Comments and Suggestions for Authors

The paper has been improved substantially, but you still need to work on the presentation of your results.

Response: Thank you very much. We greatly admire your seriousness, responsibility and meticulous spirit。

In the first and second review, you put forward many comments on narrative mode and writing ideas. Because of differences in language habits, this is a big challenge for us to revise. However, we still tried our best to revise our manuscript according to your comments. Even so, we still feel that some revised problems may not meet your requirements. Therefore, in the next review, we sincerely request you to directly tell us the revision results if there is any need to revise.

Please review it again and give your valuable comments.

Thanks again!

ABSTRACT:

(1) Your abstract has been improved substantially, but I still thinking that it needs a further effort at some points. The abstract should include a short introduction presenting the topic of the paper (1 sentence is enough). Then you should clearly state the aim of your study and the hypothesis that you want to test (this have not been included in your abstract). Then, a clear explanation of the experimental design adopted (including the number of animals included in the trial, all the assays performed, and the statistical analysis adopted) should be included. Then you should present your results and include one statement of conclusions (I think that both these sections are quite well written in your abstract). Specific comments are stated below. Please, modify your abstract accordingly and referring to the general structure described above.

Response: We have supplemented and improved the abstract section according to your comments. See the "Abstract" section in the revised text.

(2) L16: Eliminate “Our previous studies showed that”. This kind of sentences should be supported by a reference, and you should make your abstract unpersonal. If these are published data, you could take them as a matter of fact in the abstract. My personal suggestion is to change the sentence to “Micro RNA-15a (miR-15a) is closely related to intramuscular fat (IMF) deposition in chickens; however…” or something similar.

Response: We have revised them according to your comments. See the line 16 in the revised text.

(3) L18: make the abstract unpersonal. Avoid “we” (use passive form)

Response: We have revised them according to your comments. See the line 20-23 in the revised text.

(4) L18-22: the experimental design you adopted is still unclear and need a further explanation. At L 29-30 you present the results on dual-luciferase, but there is no mention to the dual-luciferase assessment in the description of the experimental design.

Response: We have supplemented and improved the experimental design according to your comments. See the line 20-27 in the revised text.

(5) L 24: Please, indicate P-values

Response: We have supplemented the P-values according to your comments. See the line 29 in the revised text.

INTRODUCTION

 (1) L 36-37. Delete the first two sentences. Unnecessary for me (personal comment).

Response: We have deleted first sentence according to your comments. But leave the second sentence as background. See the line 42 in the revised text.

(2) L54-56. Delete. Although interesting, anti-carcinogenic function of miRNA are unfocused with the aim of your study.

Response: We have deleted this sentence according to your comments. See the line 59 in the revised text.

MATHERIAL AND METHODS

(1) L 272: Move the sentence at L 267 (After “…developmental age”)

Response: We have revised them according to your comments. See the line 268-270 and 272 in the revised text.

(2) L334. Please provide the 260/280 and the RIN (RNA Integrity Number) values you found

Response: We have provided the 260/280 and the RIN values according to your comments. The results showed the range of the 260/280 ratios and RIN values of all total RNA samples. See the line 334-336 in the revised text.

RESULTS

(1) Your results have been improved substantially, but you still need to work on data presentation. Please, use the comments I provided for the first sub-section as a guide for your results. Furthermore, as a general comment, what does n=3 or n=8 means in figure captions? If they are the number of replicates, for each assay performed you need to define them in the material and methods section too.

Response: We are very sorry. As we can't quite understand your comments in the first revision, we could not make a thorough revision. In this time, we have re-improved substantially the result section according to your comments. We hope it can meet your requirements.

In addition, the n=3 in figure captions is the number of replicates. We have defined them in the material and methods section according to your comments. See the line 315-316, 321, 329, 346-347 and 370 in the revised text.

(2) L84. As I said in the previous revision, sub-titles in the results section should not be used to discuss the results. Do not anticipate the results that you found in the sub-title (i.e change “Over-expression of miR-15a promotes the differentiation of intramuscular preadipocytes from chicken breast muscle” to “Expression level of miR-15a in breast muscle and intramuscular preadipocytes” or something similar). Please, rephrase all the sub-titles in the results section accordingly.

Response: We have revised them according to your comments. See the line 88, 112 and 140 in the revised text.

(3) L 86-88. Please, rephrase the sentence to make it compliant with the results section. (i.e. “The expression level of miR-15a found in breast muscle at 14; 22 and 30 weeks of development was upregulated compared to the level found at 6 weeks (Figure 1)” or something similar)

Response: We have revised it according to your comments. See the line 89-90 in the revised text.

(4) L 88. Here you present the fold increase calculated on the expression level at 30 weeks, while statistical significances in the figure 1 are referred to the comparison between the expression found at 6 weeks and subsequent observations. Please clarify which time point you adopted as the reference time point for comparisons, and then be consistent both in results presentation and in the figure adopting the same time point to make comparisons (see following comment)

Figure 1: Please indicate which time point you used to make comparisons in figure caption too (i.e. “Fold variations of the microRNA-15a (miR-15a) expression level found in Gushi chicken breast muscle at 14; 22 and 30 weeks of development in respect with the level found at 6 weeks of development. Data are expressed…” or something similar).

Response: We have revised them according to your comments. See the line 100-102 in the revised text.

(5) L90. Delete sentence (this is not Material and Methods). Change to “In intramuscular preadipocytes the expression level of…”

Response: We have deleted this sentence and revised it according to your comments. See the line 92 in the revised text.

(6) L93. You cannot say that “miR-15a over-expression was able to significantly elevate the mRNA level of adipogenic marker genes” in the results section (keep that for discussion). I suggest you say only that miR-15a group had higher levels of PPARgamma and C/EBPalpha, and greater cholesterol and triglyceride accumulation in comparison to control group, and to support that with statistics.

Response: We have revised it according to your comments. See the line 94-97 in the revised text.

(7) L95. Same comment as above. Change to “miR-15a group showed a brighter coloration at Oil Red O staining (Figure 2)” or something similar. Figure 2(F) is unclear anyway, and you need to specify in figure caption which group correspond to which figure (footnotes at the bottom of the image are difficult to read). Finally, is there any numerical value that reflects the results of Oil Red O staining? It could be interesting to quantify the different response to the Oil Red in order to make some statistics (the latest part should be intended as a personal comment).

Response: We have revised them according to your comments. See the line 97-98 and 107-109 in the revised text. Meanwhile, we have added the relative content of intracellular Oil Red O according to your comments. See the line 97, 110 and 327 as well as Figure 2 in the revised text.

(8) L97-98. Delete. This is not the discussion section.

Response: We have deleted this sentence according to your comments. See the line 98 in the revised text.

(9) L 110-111. See previous comments and change the sub-title accordingly

Response: We have revised it according to your comments. See the line 112 in the revised text.

(10) L112-114. Both these lines, together with figure 3 (A), cannot be considered as a part of your results to me. It should be considered as a part of the introduction (it is a part of the background that drive you to the hypothesis you tested, isn’t it?). I suggest you moving this part to the introduction.

Response: We have moved this part to the introduction according to your comments. See the line 79-80 in the revised text. In addition, we have also deleted Figure 3A according to your comments. See the Figure 3 in the revised text.

(11) L114-121 Rephrase according with previous comments.

Figure 4 and 5. Do not use figure caption to present your results. See previous comments and rephrase figure caption accordingly.

Response: We have rephrased this paragraph according to your comments. See the line 113-122 in the revised text.In addition, we have also rephrased figure caption in Figure 4 and 5 accordingly. See the line 132-134 and 136-138 in the revised text.

(12) L121-122. Delete. This is not the discussion section

Response: We have deleted this sentence according to your comments. See the line 122 in the revised text.

(13) L143. Rephrase according with previous comments

Response: We have revised it according to your comments. See the line 140 in the revised text.

(14) L144-146 Delete. You already said that in the M&M section

Response: We have deleted this sentence according to your comments. See the line 141 in the revised text.

(15) Figure 6. L159. What about negative controls in figure caption?

Response: We have added negative controls in figure caption. See the line 155 in the revised text.

(16) L147-149. Rephrase according with previous comments

Response: We have revised it according to your comments. See the line 142-145 in the revised text.

(17) L149. From “Subsequently”to “cells”, Delete. See previous comments

Response: We have deleted this sentence according to your comments. See the line 145 in the revised text.

(18) Figure 6 E, F and G. Specify in caption that WT is wild type and MUT is mutate.

Response: We have added them in figure caption. See the line 161 in the revised text.

DISCUSSION

The discussion is still unfocused on your results. L 168 to 181 are a repetition of what you already said in the introduction and do not discuss any of your results. L 181 to 185 are a repetition of your results, that do not really provide any explanation to them. You should merge these two paragraphs using your results to select the information that are essential to support your interpretation of results. Please do that in the whole discussion section.

Response: We are very sorry. As we can't quite understand your comments in the first revision, we could not make a thorough revision. In this time, we have re-improved the discussion section according to your comments. We hope it can meet your requirements. See the line 165-176 and 186-195 in the revised text.
